# Throughput Maximization for UAV-Enabled Relaying in Wireless Powered Communication Networks

**DOI:** 10.3390/s19132989

**Published:** 2019-07-06

**Authors:** Yinfeng Li, Dingcheng Yang, Yu Xu, Lin Xiao, Haole Chen

**Affiliations:** 1Information Engineering School, Nanchang University, Nanchang 330031, China; 2Electronic Information School, Wuhan University, Wuhan 430072, China

**Keywords:** unmanned aerial vehicle (UAV), relaying, wireless powered communication networks (WPCN), wireless power transfer (WPT), trajectory optimization

## Abstract

This paper investigates mobile relaying in wireless powered communication networks (WPCN), where an unmanned aerial vehicle (UAV) is employed to help information delivery from multiple sources to destination with communication channels severely blocked. The sources are low-power without energy supply. To support information transmission, the UAV acts as a hybrid access point (AP) to provide wireless power transfer (WPT) and information reception for sources. We set the issue of system throughput maximization as the optimization problem. On the one hand, the system is subject to the information causality constraint due to the dependent processes of information reception and transmission for the UAV. On the other hand, the sources are constrained by a so-called neutrality constraints due to the dependent processes of energy harvesting and energy consumption. In addition, we take account of the access delay issue of all ground nodes. Specifically, two paradigms of delay-tolerant case and delay-sensitive case are presented. However, the formulated problem including optimizations for time slot scheduling, power allocation and UAV trajectory is non-convex and thus is difficult to obtain its optimal solution. To tackle this problem, we apply the successive convex approximation (SCA) technique and propose an iterative algorithm by which a suboptimal solution can be achieved. Simulation results validate our proposed design, and show that the obtained suboptimal solution is high-quality, as compared to benchmark scheme.

## 1. Introduction

Owing to their swift mobility, unmanned aerial vehicles (UAV) have been widely and increasingly applied in wireless communication [1]. At present, artificial intelligence (AI), internet of things (IoT) and big data have induced a great deal of interests in academia. How to support massive low-power wireless devices in future wireless communication system is a critical and attractive topic considering the dramatic growth of devices.

On the one hand, wireless power transfer (WPT) is considered to be a prospective technology applied to replenish power for low-power rechargeable wireless devices (WDs) by radio frequency [2,3,4,5]. In general, WPT can be divided into two applications: simultaneous wireless information and power transfer (SWIPT) and wireless powered communication networks (WPCN). In WPCN, an access point (AP) is dedicated to charge the WDs in downlink with WPT, and then the WDs can perform wireless information transfer using the harvested energy. In particular, the AP in WPCN is usually fixed at a location, which is generally faced with “doubly near–far" problem [2] or leads to the terrible efficiency if the distance between the AP and rechargeable wireless device is large. To deal with these challenges, UAV-enabled WPCN is proposed in [6,7,8]. In these works, UAV is used as a hybrid AP, namely the UAV first charges the WDs via WPT, and then receives the data from WDs via wireless information transmit (WIT), to enhance the performance of both WPT and WIT.

On the other hand, UAV-enabled mobile relaying is considered to be able to enlarge the communication coverage [9,10,11,12,13,14,15,16]. Compared with traditional static relaying, UAV-enabled relaying provides a new possibility for on-demand communication with its swift deployment/placement. Furthermore, UAV-enabled relaying can provide line-of-sight (LoS) links. By flying in proximity to the ground user, low-pathloss communication channels can be achieved. In [9], UAVs is employed to combine with wireless caching, where the UAVs are dedicated to provide content caching service for users as an airborne relay with the goal of users’ quality-of-experience maximization. In [10], significant performance gain for UAV relaying is achieved via trajectory optimization as well as power control. The authors of [11] studied both the spectrum efficiency and energy efficiency when a UAV acts as a relay, but only the circular trajectory pattern is considered. The authors of [12] maximized the UAV-enabled relaying throughput. The authors of [13] also studied throughput maximization problem in UAV-enabled relaying by joint transmit power allocation optimization and relay trajectory design. The authors of [14] studied the optimal placement of UAV relay, and obtained the altitude of both static and mobile UAV. The authors of [15] also studied the placement problem in UAV relaying networks, and optimized the transmission power, bandwidth and UAV’s position to maximize the system throughput. However, only the static UAV was considered in [15]. The problem of decoding error probability minimization was studied in UAV relaying system [16], by finding the optimal position of UAV and optimizing the block length allocation.

Given the advantages of UAV-enabled WPCN and mobile relaying, a new UAV-enabled relaying system combined with WPCN is proposed, as shown in Figure 1. In this system model, a UAV is employed as a decode-and-forward (DF) relay to assists information transmission from multiple rechargeable ground sources to a ground destination with direct links seriously blocked. In addition, the UAV plays a role as a hybrid AP, namely in each time slot it first broadcasts energy RF signal to charge all the sources via WPT, and then collects the information from the sources. We assume the destination acts as the information collection and processing node that can obtain stable power supply. Furthermore, the access delay issues for all ground nodes including sources and destination are considered, which are partitioned into two cases" delay-tolerant case and delay-sensitive case. Specifically, each source’s throughput during the finite period time or in each time slot is taken account. For convenience of adjusting the fairness or access delay among sources, we are dedicated to set a factor to balance each source’s throughput. Certainly, the access delay of destination is adjustable by another factor. In this work, we aim to achieve the throughput maximization, namely the information that successfully receives by destination from UAV or the information that successfully forwards from UAV to destination, subject to the maximum speed, information causality as well as so-called neutrality constraints [4]. To this end, we jointly optimize the time slot scheduling, power allocation and UAV’s trajectory for achieving the maximum throughput. Nevertheless, the non-convexity results in that the formulated original problem cannot be directly solved.

It is worth noting that there have been few prior studies on UAV-enabled relaying in WPCN in academia. The authors of [17] studied UAV-enabled amplify-and-forward (AF) relaying to assist information transmission from base station (BS) to destination with the goal of network throughput maximization, in which the UAV is energy-constrained and harvests energy from the BS via WPT. Evidently, our paper is significantly different from this work in scenario setup [17]. To illustrate the innovations, we summarize the main contributions of this paper as follows.

Firstly, for the mobile relaying, the UAV acted as a DF relay to help information delivery. Due to the processing delay of the DF relay, the information transmission is subject to the information causality constraint, namely the transmitted information to destination from the UAV in one time slot cannot exceed the total received information from sources before this slot.Secondly, for the WPCN, we consider multiple rechargeable sources with no residual energy at the beginning. The sources first harvest the energy from the hybrid AP (i.e., UAV) in downlink via WPT and then transmit information in uplink via WIT. In the process of WIT, the sources are restricted by neutrality constraints, i.e., each source’s energy consumption in one slot cannot exceed the total harvested energy before.Thirdly, the access delay problem is considered in this paper. For the delay-tolerant case, it is assumed that the sources and destination operate the communication protocol without strict access delay consideration in each time slot. In other words, the transmitted datas are delay-insensitive for ground nodes in delay-tolerant case. To guarantee the fairness among sources, a specific factor is set to adjust each source’s throughput over the finite period time.Next, for the delay-sensitive case, each source is requested to upload a certain amount of data to the UAV in each time slot, and the destination is also required to receive a certain amount of data forwarded by the UAV. To this end, two other specific factors are introduced to design this case.Finally, we jointly optimize time slot scheduling, power allocation and UAV trajectory to achieve a maximum throughput by applying successive convex approximation (SCA) technique. An effective iteration algorithm is proposed to solve the non-convex problem, leading to a suboptimal solution. The simulation results validate the feasibility and superiority of the proposed design. The proposed algorithm and simulation results provide helpful insights for UAV-enabled relaying design in WPCN, in which the impact of delay is considered and the UAV trajectory is optimized as well. The simulation results show that, in practice, the proposed algorithm contributes to achieve different delay required UAV-enabled WPCN system design.

## 2. System Model

A UAV-enabled WPCN is considered as illustrated in Figure 1. In this model, a UAV acts as a mobile relay that is dispatched to transmit the information from multiple sources to the destination over the area of interest. Assume that the destination has stable power supply, and the UAV also has stable power supply onboard and the equipped energy is sufficient to support the task in the finite period time *T*, but there is no power supply embedded in the sources. For information reception, the UAV needs to travel over the sources to operate WPT in downlinks, and the sources utilize the harvested energy to support WIT in uplinks. All sources are equipped with two antennas that are dedicated for energy harvesting and WIT, respectively. The UAV is also equipped with two antennas for WPT and information reception or transmission, while the destination is equipped with single antenna for information reception.

In this paper, the UAV’s location projected on the ground at time instant t∈[0,T] is denoted by ${\left[x\left(t\right),y\left(t\right)\right]}^{T}$. We use K≜{1,2,...,K} to denote the sources set. For source k∈K, its location is denoted by $w_{k}={\left[x_{k},y_{k}\right]}^{T}$, while the location of the destination is denoted by $w_{d}={\left[x_{d},y_{d}\right]}^{T}$. In addition, we assume that the UAV flies at a fixed altitude *H* and the locations of the sources and destination are fixed. For convenience of discussion, the time horizon *T* is equally discretized into *N* time slots that are represented by N≜{1,2,...,N} by using a sufficient small step $\delta_{t}$, i.e., $T=N\delta_{t}$. Thus, the trajectory of the UAV can be determined by the sequence ${\left\{q\left[n\right]={\left[x\left[n\right],y\left[n\right]\right]}^{T}\right\}}_{n=1}^{N}$, where the equality $q\left[n\right]=q\left(n\delta_{t}\right)$ holds. We assume the UAV must fly back to the initial location after period *T* for equipment maintenance or battery charge. Hence, we obtain the following constraints
(1)$$\begin{array}{c} q[1]=q[N], \end{array}$$
(2)$$\begin{array}{c} ||q\left[n+1\right]-{q\left[n\right]||}^{2}⩽{\left(V_{\max}\delta_{t}\right)}^{2},\forall n\in \overset{˘}{N}, \end{array}$$
where $\overset{˘}{N}≜\left\{1,2,...,N-1\right\}$, and $V_{\max}$ denotes the maximum speed. The expression in Equation (Equation 2) indicates the maximum travel distance of the UAV in one time slot. During the period *T*, the distance between the source *k* and the UAV and that between the UAV and the destination can be expressed as follows, respectively,(3)$$\begin{array}{c} d_{kr}\left[n\right]=\sqrt{||q\left[n\right]-w_{k}{\left|\right|}^{2}+H^{2}},k\in K,n\in N, \end{array}$$
(4)$$\begin{array}{c} d_{rd}\left[n\right]=\sqrt{||q\left[n\right]-w_{d}{\left|\right|}^{2}+H^{2}},n\in N. \end{array}$$

It is assumed that the air-to-ground/ground-to-air channel is dominated by LoS link and the Doppler effect is perfectively compensated [18,19,20,21,22,23]. Hence, we can obtain the channel power gain as
(5)$$\begin{array}{c} h_{kr}\left[n\right]=\frac{\beta_{0}}{||q\left[n\right]-w_{k}{\left|\right|}^{2}+H^{2}},k\in K,n\in N, \end{array}$$
(6)$$\begin{array}{c} h_{rd}\left[n\right]=\frac{\beta_{0}}{||q\left[n\right]-w_{d}{\left|\right|}^{2}+H^{2}},n\in N, \end{array}$$
where $h_{kr}\left[n\right]$ denotes the channel power gain from the source *k* to UAV, and $h_{rd}\left[n\right]$ is the channel power gain from the UAV to destination. In addition, $\beta_{0}$ denotes the channel power at the reference distance $d_{0}=1$ m.

In the WPCN, a DF relaying is considered based on time-division duplexing (TDD) model that equal bandwidth is allocated for information reception and forwarding. For the multiple sources, each source can share the whole frequency for information transmission through a time-division multiple access (TDMA) approach. Specifically, the communication protocol of our system is shown as Figure 2. We divide time slot n∈N into K+2 subslots that are denoted by $\overset{˘}{K}≜\left\{0\right\}\cup K\cup \left\{K+1\right\}$, and introduce an time allocation variable $\tau_{k}\left[n\right]$, with $k\in \overset{˘}{K}$, to determine the duration of the *k*-th subslot. Specifically, the 0-th slot of duration $\tau_{0}\left[n\right]\delta_{t}$ is used to operate WPT, and *k*-th slot of duration $\tau_{k}\left[n\right]\delta_{t}$ is used for WIT, with k∈K. The last subslot of duration $\tau_{K+1}\left[n\right]\delta_{t}$ belongs to the relay for forwarding the information to the destination. As a result, the time allocation variable $\tau_{k}\left[n\right]$ needs to satisfy the following constraints,
(7)$$\begin{array}{c} 0\leq \tau_{k}\left[n\right]\leq 1,k\in \overset{˘}{K},n\in N, \end{array}$$
(8)$$\begin{array}{c} \sum_{k=0}^{K+1}\tau_{k}\left[n\right]\leq 1,n\in N. \end{array}$$

In addition, it is assumed that in the first subslot of each time slot, the UAV broadcasts wireless energy signals in downlink with the constant transmit power. Thus, the harvested energy of the source *k* in time slot *n* can be expressed as    

(9)$$\begin{array}{cc} E_{k}\left[n\right] & =\tau_{0}\left[n\right]\delta_{t}\zeta_{k}h_{kr}\left[n\right]P_{E}\\ & =\frac{\tau_{0}\left[n\right]\delta_{t}\zeta_{k}\beta_{0}P_{E}}{||q\left[n\right]-w_{k}{\left|\right|}^{2}+H^{2}},k\in K,n\in N. \end{array}$$

In Equation (Equation 9), the parameter $\zeta_{k}$ denotes the energy harvesting efficiency for the source *k*. In this paper, we let $\zeta_{k}=\zeta ,\forall k\in K$, namely all the sources have the same energy harvesting efficiency. For simplicity, we consider that the source *k* transmit wireless information to the relay with a transmit power $p_{kr}\left[n\right]$, and the relay forwards the received information to the destination with the power of $p_{rd}\left[n\right]$. Assume that $p_{kr}\left[n\right]$ and $p_{rd}\left[n\right]$ satisfy the peak power and average power budget constraints, i.e.,
(10)$$\begin{array}{c} 0\leq p_{kr}\left[n\right]\leq P_{kr}^{max}, \end{array}$$
(11)$$\begin{array}{c} \frac{1}{N}\sum_{n=1}^{N}p_{rd}\left[n\right]\leq {\bar{P}}_{rd}, \end{array}$$
(12)$$\begin{array}{c} 0\leq p_{rd}\left[n\right]\leq P_{rd}^{max}, \end{array}$$
where ${\bar{P}}_{rd}$ is the average power budget constraints for the relay. $P_{kr}^{max}$ and $P_{rd}^{max}$ are the peak power that can be achieved by the sources and relay in each time slot, respectively. For the purpose of ensuring the non-trivial effects of the constraint (Equation 11), we let $P_{rd}^{max}\geq {\bar{P}}_{rd}$.

Then, the maximum transmit rate (bps/Hz) from the source *k* to the relay in time slot *n* can be written by
(13)$$\begin{array}{cc} R_{kr}\left[n\right] & =\log_{2}\left(1+\frac{h_{kr}\left[n\right]p_{kr}\left[n\right]}{\sigma^{2}}\right)\\ & =\log_{2}\left(1+\frac{\gamma_{0}p_{kr}\left[n\right]}{||q\left[n\right]-w_{k}{\left|\right|}^{2}+H^{2}}\right),k\in K,n\in N, \end{array}$$
where $\gamma_{0}=\beta_{0}/\sigma^{2}$ denotes the reference received signal-to-noise ratio (SNR) at $d_{0}=1$ m. Similarly, the achievable transmit rate of the relay or the achievable reception rate of the destination in slot *n* is written by

(14)$$\begin{array}{cc} R_{rd}\left[n\right] & =\log_{2}\left(1+\frac{h_{rd}\left[n\right]p_{rd}\left[n\right]}{\sigma^{2}}\right)\\ & =\log_{2}\left(1+\frac{\gamma_{0}p_{rd}\left[n\right]}{||q\left[n\right]-w_{d}{\left|\right|}^{2}+H^{2}}\right),n\in N, \end{array}$$

It is worth noting that, due to the processing delay of the energy circuits of the sources, the harvested energy in time slot *n* can only be available in future time slot m>n, with m∈N. Therefore, for each source, we obtain the energy neutrality constraint as follows,
(15)$$\begin{array}{c} \sum_{j=2}^{n}\tau_{k}\left[j\right]\delta_{t}p_{kr}\left[n\right]\leq \sum_{j=1}^{n-1}E_{k}\left[j\right],k\in K,n\in \overset{˘}{N}, \end{array}$$
where $\overset{˘}{N}≜\left\{2,3,...,N\right\}$. Furthermore, the inequality in Equation (Equation 15) indicates the 1st time slot just for WPT, i.e., $\tau_{0}\left[1\right]=1$, and the *N*-th time slot do not operate WPT, i.e., $\tau_{0}\left[N\right]=0$.

Next, the information causality is considered because the processing delay incurred by DF relaying is nonegligible. For simplicity, we adopt one time slot to represent the processing delay of the UAV. As a result, the information causality constraint is given by
(16)$$\begin{array}{c} \sum_{j=3}^{n}\tau_{K+1}\left[j\right]R_{rd}\left[j\right]\leq \sum_{k=1}^{K}\sum_{j=2}^{n-1}\tau_{k}\left[j\right]R_{kr}\left[j\right],n\in \hat{N}, \end{array}$$
where $\hat{N}≜\left\{3,4,...,N\right\}$. Here, we give a brief explanation for Equation (Equation 16). First, from the above statement, we know the first and second time slots are allocated for WPT and information uploading, respectively. Namely, the information transmit for the relay starts from the third slot. In addition, due to the assumption of processing delay for the relay, resulting in that the last time slot, i.e., the *N*th slot, cannot be used to operate WIT for sources, otherwise it would cause time resource waste. The information causality indicates that the WIT starts from the second time slot and the *N*th slot can only be used for information forwarding to the destination, i.e., $\tau_{K+1}\left[N\right]=1$. It can be easily proved that, at the optimal solution, the inequality in Equation (Equation 16) would hold with equality. Otherwise, we can always increase the value of $\tau_{K+1}\left[j\right]R_{rd}\left[j\right]$ by enlarging the value of $\tau_{K+1}\left[j\right]$, which does not violate our design.

In delay-tolerant case, we use a factor $\theta_{k}\in \left[0,1\right]$ to stand for the required throughput of the source *k* over the average achievable throughput of all sources in period time *T*, i.e.,

(17)$$\begin{array}{c} \sum_{n=2}^{N}\tau_{k}\left[n\right]R_{kr}\left[n\right]\geq \theta_{k}\frac{1}{K}\sum_{k=1}^{K}\sum_{n=2}^{N}\tau_{k}\left[n\right]R_{kr}\left[n\right],\forall k\in K. \end{array}$$

While in the delay-sensitive case, each source is requested to upload a certain amount of data to the UAV in each time slot, and simultaneously the destination also requires receiving a certain amount of data forwarded by the UAV. To this end, we introduce two factors of $\lambda_{k}\in \left[0,1\right]$ and υ∈[0,1] to indicate the ratio of required throughput of the source *k* and destination in per time slot over the average achieved throughput of sources and destination, respectively, which are shown in Equations (Equation 18) and (Equation 19),
(18)$$\begin{array}{c} \tau_{k}\left[n\right]R_{kr}\left[n\right]\geq \frac{\lambda_{k}}{KN}\sum_{k=1}^{K}\sum_{n=2}^{N}\tau_{k}\left[n\right]R_{kr}\left[n\right],\forall k\in K,n\in \overset{˘}{N}. \end{array}$$
(19)$$\begin{array}{c} \tau_{K+1}\left[n\right]R_{rd}\left[n\right]\geq \frac{\upsilon }{N}\sum_{n=3}^{N}\tau_{K+1}\left[n\right]R_{rd}\left[n\right],n\in \hat{N}. \end{array}$$

In the system, we intend to maximize the achievable throughput of the destination by taking account of the optimizations of scheduling, power allocation and UAV trajectory. Thus, for the delay-tolerant case, the problem is given by
$$\begin{array}{cc} \left(P1\right) & \max_{\tau_{k}\left[n\right],p_{kr}\left[n\right],p_{rd}\left[n\right],q\left[n\right]}\frac{1}{N}\sum_{n=3}^{N}\tau_{K+1}\left[n\right]R_{rd}\left[n\right]\\ s.t.\;\; & (1),(2),(7),(8),(10)–(12),(15)–(17). \end{array}$$

As for the delay-sensitive case, the corresponding problem can be given by$$\begin{array}{cc} \left(P2\right) & \max_{\tau_{k}\left[n\right],p_{kr}\left[n\right],p_{rd}\left[n\right],q\left[n\right]}\frac{1}{N}\sum_{n=3}^{N}\tau_{K+1}\left[n\right]R_{rd}\left[n\right]\\ s.t.\;\; & (1),(2),(7),(8),(10)–(12),(15),(16),(18),(19). \end{array}$$

Note that Problem (P1) is the same as Problem (P2) except for the constraints (Equation 17)–(Equation 19), while actually the expressions in Equations (Equation 17) and (Equation 18) are similar, and the constraint (Equation 19) is also related to the objective function of Problem (P1), which inspires us to find a solution to Problem (P1) that can be equivalently applied to solve Problem (P2). However, Problem (P1) is a non-convex problem due to the non-convex objective function and the non-convexity in Equations (Equation 15)–(Equation 17). To obtain the solution of this problem, an effective iteration algorithm is designed, with which a suboptimal solution is achieved in the Section 3.

## 3. Proposed Design

To solve Problem (P1), the original problem can be considered as one consisting of three subproblems. The first subproblem is to optimize time slot scheduling $\left\{\tau_{k}\left[n\right]\right\}$, the second subproblem is to optimize power allocation $\left\{p_{kr}\left[n\right],p_{rd}\left[n\right]\right\}$, and UAV trajectory {q[n]} optimization is the last subproblem. Given the other variables, each subproblem can be well tackled. Based on this decomposition, we propose an effective iterative algorithm by alternately optimizing these three subproblems until they converge to a locally optimal value.

### 3.1. Time Slot Scheduling Optimization

In this subsection, the time slot scheduling $\left\{\tau_{k}\left[n\right]\right\}$ is optimized by given power allocation $\left\{p_{kr}^{\left(i\right)}\left[n\right],p_{rd}^{\left(i\right)}\left[n\right]\right\}$ and UAV trajectory $\left\{q^{\left(i\right)}\left[n\right]\right\}$, with *i* denoting *i*-th iteration. Thus, the subproblem can be formulated as
$$\begin{array}{cc} \left(P1.1\right)\max_{\tau_{k}\left[n\right]} & \sum_{n=3}^{N}\tau_{K+1}\left[n\right]R_{rd}\left[n\right]\\ s.t.\;\; & (7),(8),(15)–(17). \end{array}$$

Note that we neglect the constant $\frac{1}{N}$ in objective function of Problem (P1.1) and other deduced problems hereinafter. Problem (P1.1) is a convex problem, as Equations (Equation 7), (Equation 8), and (Equation 15)–(Equation 17) are affine with respect to $\left\{\tau_{k}\left[n\right]\right\}$, and the objective function is linear. Thus, it can be easily solved by using linear programming (LP) methods or convex optimization tools such as CVX [24]. For convenience of notation, let $\left\{\tau_{k}^{\left(i+1\right)}\left[n\right]\right\}$ denote the optimized solution for Problem (P1.1).

### 3.2. Power Allocation Optimization

For given $\left\{q^{\left(i\right)}\left[n\right]\right\}$ and $\left\{\tau_{k}^{\left(i+1\right)}\left[n\right]\right\}$, the optimization problem of power allocation is formulated as
$$\begin{array}{cc} \left(P1.2\right)\max_{p_{kr}\left[n\right],p_{rd}\left[n\right]} & \sum_{n=3}^{N}\tau_{K+1}^{\left(i+1\right)}\left[n\right]R_{rd}\left[n\right]\\ s.t.\;\; & (10)–(12),(15)–(17). \end{array}$$

Obviously, the non-convexity in the constraints in constraints (Equation 16) and (Equation 17) makes Problem (P1.2) non-convex. For further processing, introducing nonnegative slack variables $\left\{{\hat{g}}_{k}\left[n\right]\right\}$ and $\left\{\overset{ˇ}{g}\left[n\right]\right\}$ that meet the following constraints, respectively,
(20)$$\begin{array}{c} \hat{g}\left[n\right]\leq R_{rd}\left[n\right],\forall n\in N, \end{array}$$
(21)$$\begin{array}{c} {\tilde{g}}_{k}\left[n\right]\leq R_{kr}\left[n\right],\forall k\in K,n\in N. \end{array}$$

The obtained constraints in constraints (Equation 20) and (Equation 21) are convex. As a result, Problem (P1.2) can be converted into a new form as below
$$\begin{array}{cc} & \left(P1.2.1\right)\max_{p_{kr}\left[n\right],p_{rd}\left[n\right],\hat{g}\left[n\right],{\overset{ˇ}{g}}_{k}\left[n\right]}\sum_{n=3}^{N}\tau_{K+1}^{\left(i+1\right)}\left[n\right]\hat{g}\left[n\right]\\ s.t.\;\; & (10)–(12),(15), \end{array}$$
(22)$$\sum_{j=3}^{n}\tau_{K+1}^{\left(i+1\right)}\left[j\right]\hat{g}\left[j\right]\leq \sum_{k=1}^{K}\sum_{j=2}^{n-1}\tau_{k}^{\left(i+1\right)}\left[j\right]{\overset{ˇ}{g}}_{k}\left[j\right],n\in \hat{N},$$
(23)$$\begin{array}{c} \sum_{n=2}^{N}\tau_{k}^{\left(i+1\right)}\left[n\right]{\overset{ˇ}{g}}_{k}\left[n\right]\geq \theta_{k}\frac{1}{K}\sum_{k=1}^{K}\sum_{n=2}^{N}\tau_{k}^{\left(i+1\right)}\left[n\right]{\overset{ˇ}{g}}_{k}\left[n\right],\forall k\in K. \end{array}$$
(24)$$\begin{array}{c} \hat{g}\left[n\right]\leq \log_{2}\left(1+\alpha_{n}p_{rd}\left[n\right]\right),n\in N, \end{array}$$
(25)$$\begin{array}{c} {\overset{ˇ}{g}}_{k}\left[n\right]\leq \log_{2}\left(1+\beta_{n}p_{kr}\left[n\right]\right),k\in K,n\in N, \end{array}$$
(26)$$\begin{array}{c} \hat{g}\left[n\right]\geq 0,n\in N, \end{array}$$
(27)$$\begin{array}{c} {\overset{ˇ}{g}}_{k}\left[n\right]\geq 0,k\in K,n\in N, \end{array}$$
where $\alpha_{n}=\frac{\gamma_{0}}{||q\left[n\right]-w_{d}{\left|\right|}^{2}+H^{2}}$ and $\beta_{n}=\frac{\gamma_{0}}{||q\left[n\right]-w_{k}{\left|\right|}^{2}+H^{2}}$. It can be easily proved that the inequalities in constraints (Equation 24) and (Equation 25) must be satisfied with equalities when the optimal solution is achieved for Problem (P1.2.1). Otherwise, the value of the objective function is always nondecreasing by increasing the value of $\left\{{\hat{g}}_{k}\left[n\right]\right\}$ and $\left\{\overset{ˇ}{g}\left[n\right]\right\}$. Hence, (P1.2.1) and (P1.2) are equivalent, which means that these two problems share the same optimal solution.

By analyzing the above, by solving the approximately convex Problem (P1.2.1) of Problem (P1.2) using CVX, the optimal power allocation, denoted by $\left\{p_{kr}^{\left(i+1\right)}\left[n\right],p_{rd}^{\left(i+1\right)}\left[n\right]\right\}$, can be achieved.

### 3.3. UAV Trajectory Optimization

In this section, we optimize the UAV trajectory {q[n]} with given $\tau_{k}^{\left(i+1\right)}\left[n\right]$ and $\left\{p_{kr}^{\left(i+1\right)}\left[n\right],p_{rd}^{\left(i+1\right)}\left[n\right]\right\}$. Based on Problem (P1), this subproblem can be formulated as
$$\begin{array}{cc} \left(P1.3\right)\; & \max_{q[n]}\sum_{n=3}^{N}\tau_{K+1}^{\left(i+1\right)}\left[n\right]R_{rd}\left[n\right]\\ s.t.\;\; & (1),(2),(15)–(17). \end{array}$$

Obviously, Problem (P1.3) is non-convex caused by the constraints in constraints in Equations (Equation 15)–(Equation 17) as well as the objective function. We first consider introducing slack variables $\left\{u_{k}\left[n\right]\right\}$ and $\left\{u_{d}\left[n\right]\right\}$ that satisfy $u_{k}\left[n\right]\geq ||q\left[n\right]-w_{k}{\left|\right|}^{2}+H^{2}$ and $u_{d}\left[n\right]\geq ||q\left[n\right]-w_{d}{\left|\right|}^{2}+H^{2}$, respectively. In addition, we define $z_{E_{k},j}^{\left(i+1\right)}≜\tau_{0}^{\left(i+1\right)}\left[j\right]\zeta_{k}\beta_{0}P_{E}$. Thus, Problem (P1.3) is given by
$$\begin{array}{cc} & \left(P1.3.1\right)\max_{q\left[n\right],u_{k}\left[n\right],u_{d}\left[n\right]}\sum_{n=3}^{N}\tau_{K+1}^{\left(i+1\right)}\left[n\right]\log_{2}\left(1+\frac{\gamma_{0}p_{rd}^{\left(i+1\right)}\left[n\right]}{u_{d}\left[n\right]}\right)\\ s.t.\;\; & (1),(2), \end{array}$$
(28)$$\sum_{j=2}^{n}\tau_{k}^{\left(i+1\right)}\left[j\right]p_{kr}^{\left(i+1\right)}\left[j\right]\leq \sum_{j=1}^{n-1}\frac{z_{E_{k},j}^{\left(i+1\right)}}{u_{k}\left[j\right]},k\in K,n\in \overset{˘}{N},$$
(29)$$\begin{array}{c} \sum_{j=3}^{n}\tau_{K+1}^{\left(i+1\right)}\left[j\right]\log_{2}\left(1+\frac{\gamma_{0}p_{rd}^{\left(i+1\right)}\left[j\right]}{u_{d}\left[j\right]}\right)\leq \sum_{k=1}^{K}\sum_{j=2}^{n-1}\tau_{k}^{\left(i+1\right)}\left[j\right]\log_{2}\left(1+\frac{\gamma_{0}p_{kr}^{\left(i+1\right)}\left[j\right]}{u_{k}\left[j\right]}\right),n\in \hat{N}, \end{array}$$
(30)$$\begin{array}{c} \sum_{n=2}^{N}\tau_{k}^{\left(i+1\right)}\left[n\right]\log_{2}\left(1+\frac{\gamma_{0}p_{kr}^{\left(i+1\right)}\left[n\right]}{u_{k}\left[n\right]}\right)\geq \theta_{k}\frac{1}{K}\sum_{k=1}^{K}\sum_{n=2}^{N}\tau_{k}^{\left(i+1\right)}\left[n\right]\log_{2}\left(1+\frac{\gamma_{0}p_{kr}^{\left(i+1\right)}\left[n\right]}{u_{k}\left[n\right]}\right),\forall k\in K, \end{array}$$
(31)$$\begin{array}{c} u_{k}\left[n\right]\geq ||q\left[n\right]-w_{k}{\left|\right|}^{2}+H^{2},\forall n\in \overset{˘}{N}, \end{array}$$
(32)$$\begin{array}{c} u_{d}\left[n\right]\geq ||q\left[n\right]-w_{d}{\left|\right|}^{2}+H^{2},\forall n\in \hat{N}. \end{array}$$

It can be verified that the equality must hold in both Equations (Equation 31) and (Equation 32) at the optimal solution to Problem (P1.3.1), otherwise we can also increase the objective value by decreasing the value of $\left\{u_{d}\left[n\right]\right\}$, or enlarge the upper bound of the objective value by decreasing the value of $\left\{u_{k}\left[n\right]\right\}$, as shown in Equation (Equation 29). Therefore, Problem (P1.3.1) is equivalent to Problem (P1.3). Nevertheless, we still need to tackle the non-convex constraints in Equations (Equation 28)–(Equation 30) and the objective function to make Problem (P1.3.1) tractable. To deal with the non-convexity of the objective function in Problem (P1.3.1), we introduce the slack variables {s[n]} that satisfy the conditions as follows,
(33)$$\begin{array}{c} 0\leq s\left[n\right]\leq \log_{2}\left(1+\frac{\gamma_{0}p_{rd}^{\left(i+1\right)}\left[n\right]}{u_{d}\left[n\right]}\right), \end{array}$$

Then we employ SCA technique to deal with the the constraint in (Equation 33). For given feasible $u_{d}^{\left(i\right)}\left[n\right]$, the lower bound function of $\log_{2}\left(1+\frac{\gamma_{0}p_{rd}^{\left(i+1\right)}\left[n\right]}{u_{d}\left[n\right]}\right)$ via its first-order first-order Taylor expansion with respect to $u_{d}\left[n\right]$ can be obtained as,
(34)$$\begin{array}{cc} \log_{2}\left(1+\frac{\gamma_{0}p_{rd}^{\left(i+1\right)}\left[n\right]}{u_{d}\left[n\right]}\right) & \geq \log_{2}\left(1+\frac{\gamma_{0}p_{rd}^{\left(i+1\right)}\left[n\right]}{u_{d}^{\left(i\right)}\left[n\right]}\right)-\frac{1}{\ln2}\frac{\gamma_{0}p_{rd}^{\left(i+1\right)}\left[n\right]\left(u_{d}\left[n\right]-u_{d}^{\left(i\right)}\left[n\right]\right)}{u_{d}^{\left(i\right)}\left[n\right]\left(u_{d}^{\left(i\right)}\left[n\right]+\gamma_{0}p_{rd}^{\left(i+1\right)}\left[n\right]\right)}\\ & ≜\varphi_{rd}^{LB}\left[n\right]. \end{array}$$

The obtained function $\varphi_{rd}^{LB}\left[n\right]$ is convex with regard to $u_{d}\left[n\right]$. Similarly, the lower bound function of $\log_{2}\left(1+\frac{\gamma_{0}p_{kr}^{\left(i+1\right)}\left[n\right]}{u_{k}\left[j\right]}\right)$ in Equations (Equation 29) and (Equation 30) can be expressed as
(35)$$\begin{array}{cc} \log_{2}\left(1+\frac{\gamma_{0}p_{kr}^{\left(i+1\right)}\left[n\right]}{u_{k}\left[j\right]}\right) & \geq \log_{2}\left(1+\frac{\gamma_{0}p_{kr}^{\left(i+1\right)}\left[n\right]}{u_{k}^{\left(i\right)}\left[j\right]}\right)-\frac{1}{\ln2}\frac{\gamma_{0}p_{kr}^{\left(i+1\right)}\left[n\right]\left(u_{k}\left[j\right]-u_{k}^{\left(i\right)}\left[j\right]\right)}{u_{k}^{\left(i\right)}\left[j\right]\left(u_{k}^{\left(i\right)}\left[j\right]+\gamma_{0}p_{kr}^{\left(i+1\right)}\left[j\right]\right)}\\ & ≜\varphi_{kr}^{LB}\left[j\right]. \end{array}$$
where $\left\{u_{k}^{\left(i\right)}\left[j\right]\right\}$ denotes given feasible points at the *i*-th iteration.

Next, note that the non-convexity of the constraint in Equation (Equation 28) is determined by the term of $\frac{z_{E_{k},j}^{\left(i+1\right)}}{u_{k}\left[j\right]}$, but we know that it is convex with respect to $u_{k}\left[j\right]$. With given $\left\{u_{k}^{\left(i\right)}\left[j\right]\right\}$, we can also have
(36)$$\begin{array}{c} \frac{z_{E_{k},j}^{\left(i+1\right)}}{u_{k}\left[j\right]}\geq \frac{z_{E_{k},j}^{\left(i+1\right)}}{u_{k}^{\left(i\right)}\left[j\right]}-\frac{z_{E_{k},j}^{\left(i+1\right)}}{{\left(u_{k}^{\left(i\right)}\left[j\right]\right)}^{2}}\left(u_{k}\left[j\right]-u_{k}^{\left(i\right)}\left[j\right]\right)≜\varphi_{E_{k}}^{LB}\left[j\right]. \end{array}$$

With Equations (Equation 33)–(Equation 36), the problem (P1.3.1) can be converted into a new form as
$$\begin{array}{cc} \left(P1.3.2\right) & \max_{q\left[n\right],u_{k}\left[n\right],u_{d}\left[n\right],s\left[n\right]}\sum_{n=3}^{N}\tau_{K+1}^{\left(i+1\right)}\left[n\right]s\left[n\right]\\ s.t.\;\; & (1),(2),(31),(32), \end{array}$$
(37)$$\begin{array}{c} 0\leq s\left[n\right],n\in \hat{N}, \end{array}$$
(38)$$\begin{array}{c} s\left[n\right]\leq \varphi_{rd}^{LB}\left[n\right],n\in \hat{N}, \end{array}$$
(39)$$\begin{array}{c} \sum_{j=2}^{n}\tau_{k}^{\left(i+1\right)}\left[j\right]p_{kr}^{\left(i+1\right)}\left[n\right]\leq \sum_{j=1}^{n-1}\varphi_{E_{k}}^{LB}\left[j\right],k\in K,n\in \overset{˘}{N}, \end{array}$$
(40)$$\begin{array}{c} \sum_{j=3}^{n}\tau_{K+1}^{\left(i+1\right)}\left[j\right]s\left[j\right]\leq \sum_{k=1}^{K}\sum_{j=2}^{n-1}\tau_{k}^{\left(i+1\right)}\left[j\right]\varphi_{kr}^{LB}\left[j\right],n\in \hat{N}, \end{array}$$
(41)$$\begin{array}{c} \sum_{n=2}^{N}\tau_{k}^{\left(i+1\right)}\left[n\right]\varphi_{kr}^{LB}\left[n\right]\geq \theta_{k}\frac{1}{K}\sum_{k=1}^{K}\sum_{n=2}^{N}\tau_{k}^{\left(i+1\right)}\left[n\right]\varphi_{kr}^{LB}\left[n\right],\forall k\in K, \end{array}$$

It can be verified that, for the optimal solution to Problem (P1.3.2), the inequality $s\left[n\right]\leq \varphi_{rd}^{LB}\left[n\right]$ in Equation (Equation 38) must be satisfied with equality, otherwise we can increasing the value of objective function by increasing the value of s[n]. Similarly, it also can be easily verified that the inequalities in Equations (Equation 34)–(Equation 36) hold with equalities for the optimal solution. Thus, Problem (P1.3.2) is equivalent to Problem (P1.3.1), hence also Problem (P1.3). For convex Problem (P1.3.2), it can be effectively solved by CVX. Let $\left\{q^{\left(i+1\right)}\left[n\right]\right\}$ denote the optimized solution of Problem (P1.3.2). In the next iteration, the obtained $\left\{q^{\left(i+1\right)}\left[n\right]\right\}$ would be new given points to optimize Problem (P1.1).

Note that, as for the delay-sensitive case, i.e., Problem (P2), it can be solved similarly by substituting Equation (Equation 17) with Equations (Equation 18) and (Equation 19), and the non-convexity in Equations (Equation 18) and (Equation 19) are tractable using the obtained lower-bounded expressions above. In other words, the solution to Problem (P1) can also be readily applied to solve Problem (P2). Hence, we can solve the both delay-tolerant case and delay-sensitive case by a same set of procedures, which is summarized into an efficient algorithm presented in the next subsection.

### 3.4. Proposed Algorithm

Based on the analysis above, an effective iterative algorithm can be proposed via alternatively optimizing these three subproblems. Via this algorithm, at least a locally optimal solution to the original Problem (P1) or Problem (P2) can be obtained.

### 3.5. Convergence and Complexity Analysis of Proposed Algorithm

To explain the convergence of the proposed design, we use G(τ,P,Q), $G_{\tau }\left(\tau ,P,Q\right)$, $G_{P}\left(\tau ,P,Q\right)$ and $G_{Q}\left(\tau ,P,Q\right)$ as objective values of Problems (P1), (P1.1), (P1.2.1) and (P1.3.2), respectively, where $\tau ≜\left\{\tau_{k}\left[n\right]\right\},P≜\left\{p_{kr}\left[n\right],p_{rd}\right\},Q≜\left\{q\left[n\right]\right\}$. For any one iteration i>0, we can get the following expressions,
(42)$$\begin{array}{cc} G(\tau^{\left(i\right)},P^{\left(i\right)},Q^{\left(i\right)}) & \leq G_{\tau }\left(\tau^{\left(i+1\right)},P^{\left(i\right)},Q^{\left(i\right)}\right) \end{array}$$
(43)$$\begin{array}{cc} & =G(\tau^{\left(i+1\right)},P^{\left(i\right)},Q^{\left(i\right)}), \end{array}$$
since $\tau^{\left(i+1\right)}$ is the globally optimal solution to Problem (P1) by solving Problem (P1.1) with given $P^{\left(i\right)}$ and $Q^{\left(i\right)}$. Considering Problem (P1.2) always offers a lower-bounded solution to Problem (P1), we can obtain
(44)$$\begin{array}{cc} G(\tau^{\left(i+1\right)},P^{\left(i\right)},Q^{\left(i\right)}) & =G_{P}\left(\tau^{\left(i+1\right)},P^{\left(i\right)},Q^{\left(i\right)}\right) \end{array}$$
(45)$$\begin{array}{cc} & \leq G_{P}\left(\tau^{\left(i+1\right)},P^{\left(i+1\right)},Q^{\left(i\right)}\right) \end{array}$$
(46)$$\begin{array}{cc} & \leq G(\tau^{\left(i+1\right)},P^{\left(i+1\right)},Q^{\left(i\right)}), \end{array}$$
the equality Equation (Equation 44) holds because Equations (Equation 20) and (Equation 21) are tight bounds for Problem (P1). Similarly, it can be easily obtained

(47)$$\begin{array}{ccc} & G(\tau^{\left(i+1\right)},P^{\left(i+1\right)},Q^{\left(i\right)}) & =G_{Q}\left(\tau^{\left(i+1\right)},P^{\left(i+1\right)},Q^{\left(i\right)}\right) \end{array}$$

(48)$$\begin{array}{cc} & \leq G_{Q}\left(\tau^{\left(i+1\right)},P^{\left(i+1\right)},Q^{\left(i+1\right)}\right) \end{array}$$

(49)$$\begin{array}{cc} & \leq G(\tau^{\left(i+1\right)},P^{\left(i+1\right)},Q^{\left(i+1\right)}) \end{array}$$

Based on Equations (Equation 42)–(Equation 49), we have
(50)$$\begin{array}{c} G\left(\tau^{\left(i\right)},P^{\left(i\right)},Q^{\left(i\right)}\right)\leq G\left(\tau^{\left(i+1\right)},P^{\left(i+1\right)},Q^{\left(i+1\right)}\right) \end{array}$$

As a result, the proposed algorithm is verified to be non-decreasing in iterations. In addition, the objective value of Problem (P1) is upper-bounded, which means that the algorithm is guaranteed to converge.

In the proposed Algorithm 1, the original Problem (P1) is decomposed into three subproblems that can be efficiently solved by typical method, as applied in [18,20,23] with low complexity. Then, these subproblems are optimized in an alternate manner. Furthermore, the optimization tool CVX is high-efficiency for solve the such convex problems [24], which makes the complexity of Algorithm 1 affordable.

**Algorithm 1** An iterative algorithm for solving Problems (P1) and (P2)
1:Initialize power allocation $\left\{p_{kr}^{\left(i\right)}\left[n\right],p_{rd}^{\left(i\right)}\left[n\right]\right\}$ and UAV trajectory $\left\{q^{\left(i\right)}\left[n\right]\right\}$, slack variables $\left\{u_{k}^{\left(i\right)}\left[n\right]\right\}$, $\left\{u_{d}^{\left(i\right)}\left[n\right]\right\}$. Set iteration i=0, the precision ε>0.2:
**repeat**
3:    Solve subproblem (P1.1) with $\left\{p_{kr}^{\left(i\right)}\left[n\right],p_{rd}^{\left(i\right)}\left[n\right]\right\}$ and $\left\{q^{\left(i\right)}\left[n\right]\right\}$, obtain the optimized result denoted by $\left\{\tau_{k}^{\left(i+1\right)}\left[n\right]\right\}$.4:    Solve subproblem (P1.2.1) with $\left\{\tau_{k}^{\left(i+1\right)}\left[n\right]\right\}$ and $\left\{q^{\left(i\right)}\left[n\right]\right\}$, obtain the optimized result denoted by $\left\{p_{kr}^{\left(i+1\right)}\left[n\right],p_{rd}^{\left(i+1\right)}\left[n\right]\right\}$.5:    Solve subproblem (P1.3.2) with $\left\{q^{\left(i\right)}\left[n\right]\right\}$, $\left\{\tau_{k}^{\left(i+1\right)}\left[n\right]\right\}$ and $\left\{p_{kr}^{\left(i+1\right)}\left[n\right],p_{rd}^{\left(i+1\right)}\left[n\right]\right\}$ as well as $\left\{u_{k}^{\left(i\right)}\left[n\right],u_{d}^{\left(i\right)}\left[n\right]\right\}$, obtain the optimized result denoted by $\left\{q^{\left(i+1\right)}\left[n\right],u_{k}^{\left(i+1\right)}\left[n\right],u_{d}^{\left(i+1\right)}\left[n\right]\right\}$.6:    Update i=i+1.7:
**until**
8:    The algorithm converges to ε.


## 4. Numerical Results

In this section, we present the numerical results to evaluate and validate Algorithm 1, as compared to the benchmark. We considered three source nodes in the area of interest, i.e., K=3. The UAV’s altitude was fixed at H=10 m. The transmit power of the UAV for WPT was set as $P_{E}=10$ w, and the sources’ peak power was set as $P_{kr}^{max}=-20$ dBm. The UAV’s average power budget and peak power were set as ${\bar{P}}_{rd}=-25$ dBm and $P_{rd}^{max}=-20$ dBm, respectively. In addition, we set the energy harvesting efficiency as ζ=0.55, and the length of time slot as $\delta_{t}=1$ s. The communication bandwidth is B=1 MHz. Furthermore, we set $\sigma^{2}=-90$ dBm and $\beta_{0}=-30$ dB. The maximum speed of the UAV was set as $V_{\max}=3$ m/s. For convenience of expression, we use vectors $\theta \in R^{1\times K}$ and $\lambda \in R^{1\times K}$ to present the factor values of multiple sources for delay-tolerant case and delay-sensitive case, respectively, with $\theta_{k}\in \theta$ denoting the required throughput of the source *k* over the average system throughput throughout the whole period time, and $\lambda_{k}\in \lambda$ denoting the required throughput of the source *k* over the average system throughput in each time slot. For illustrating the location changes of the UAV, the trajectory is marked by “◯”s every 2 s. Meanwhile, the initial/final location of the UAV is marked by “□”s, as shown in Figure 3 and Figure 4.

In Figure 3, we illustrate the UAV trajectory of delay-tolerant case under different factor θ for a given period T=60 s. The UAV trajectory is varying with the different values of θ. It can be readily comprehended that the UAV must satisfy each source requirement for information throughput in terms of θ, as shown in Equation (Equation 17). In other words, the UAV has to change its trajectory as the cost for the purpose of achieving the required fairness.

In Figure 4, the UAV trajectory of delay-sensitive case under different factor λ and υ for a given period T=60 s. Due to the delay-sensitive design for the required throughput in each time slot, the trajectories under this case are more restricted compared to those under delay-tolerant case. Specifically, in Figure 4b, the UAV hovers in place or creeps at a small scale because all sources desire to be served to the greatest extent, thus the UAV has to sacrifice the mobility to meet the requirements of the sources.

Figure 5 shows the system scheduling along the trajectory over the period T=60 s under different cases. Specifically, we use “▵”, “◯”, and “∇” to signify energy harvesting (denoted by EH), information transmit from sources to UAV (denoted by S2U) and data forwarding from UAV to destination (denoted by U2D), respectively. Figure 5a shows that the executions of S2U and U2D are separated in many time slots. This is reasonable for delay-tolerant case, as the system scheduling is not severely limited in each time slot to satisfy the average throughput requirement over *T*. This gives more degree of freedom to the UAV as well as the sources for the utility of each time slot. For the delay-sensitive case, i.e., in Figure 5b, the scheduling is overlapped in most time slots because of the causality constraints in Equations (Equation 15) and (Equation 16), which means that each source and the UAV must be scheduled for communication in each time slot, except for the first, second and last time slots, in order to satisfy the delay-sensitive factor requirement of λ=(1,1,1),υ=1.

Figure 6 shows the specified time proportion of scheduling in each time slot for delay-tolerant case with θ=(1,1,1) and delay-sensitive case with λ=(1,1,1),υ=1 under period T=60 s. The scheduling in each slot is sorted into three operations, i.e., EH, S2U and U2D, as illustrated in Figure 5. As shown in the picture, EH is first operated as some sources have no residual power at the beginning. For the delay-tolerant case (i.e, Figure 6a), both the information uploading and information forwarding executions depend on the channel quality in each time slot. In other words, the UAV tends to communicate with the nearest node for enjoying larger information transmit rate. For the delay-sensitive case (i.e, Figure 6b), due to the severe delay constraint in each time slot, the UAV must meet certain information transmit requirements from sources to destination in each time slot. Therefore, the executions of EH, S2U and U2D are always scheduled.

Figure 7 and Figure 8 show the UAV trajectory under different period time *T* for delay-tolerant case and delay-sensitive case, respectively. In Figure 7 (i.e, the delay-tolerant case), it is observed that, with the increase of the value of *T*, the UAV tries to fly closer to sources to access better channels, which results in much wireless energy transfer as well as information reception. Meanwhile, the UAV tends to fly in close proximity to the destination for larger forwarding rate. For a lower value of *T*, the UAV is limited in terms of coverage because of the maximum mobility restriction in Equation (Equation 2). With the increasing of the value of *T*, the UAV obtains much more freedom to exploit its mobility, and even hovers over each ground user. In the delay-sensitive case (i.e., Figure 8), the UAV’s trajectories are more intensive, which indicates that the degree of freedom achieved by the UAV for trajectory design is limited to meet the delay requirement in each time slot.

Figure 9 shows the convergence performance of Algorithm 1 applied in delay-tolerant case and delay-sensitive case. It can be observed that the proposed algorithm is always non-decreasing and can quickly converge to an optimized point for different factors, which indicates that the proposed design in this paper that uses SCA technique to solve the formulated non-convex problem is effective.

To show the effectiveness of Algorithm 1, Figure 10 shows the information rate received by the destination versus the fixed energy harvesting duration in each time slot, i.e., $\tau_{0}\left[n\right]$, for given period T=60 s. For the convenience of comparison, we plot the achievable rate of proposed design in which the value of $\tau_{0}\left[n\right]$ is optimized. It can be observed that the case of fixed $\tau_{0}\left[n\right]$ is worse than the proposed design, and the delay-tolerant case always outperforms the delay-sensitive case. Specifically, for the delay-tolerant case of θ=(0,0,0), the achievable rate decreases as the value of $\tau_{0}\left[n\right]$ increases, while, for the delay-sensitive case, the achievable rate first increases and then decreases. This phenomenon can be well comprehended by combining Figure 6 and Figure 10; the operation of WPT in each time slot is not the best choice for delay-tolerant case but is for the delay-sensitive case. In general, with the increasing value of $\tau_{0}\left[n\right]$, less time can be utilized for information transmit in each time slot, thus resulting in the rate decrease for large values of $\tau_{0}\left[n\right]$.

In Figure 11, we plot the throughput of each source under different factor θ of delay-tolerant case. In this picture, it can be intuitively observed that the different factors bring significant influence to the sources’ throughput, e.g., by comparing the case of θ=(1.4,0.2,1.4) with the case of θ=(1.4,1.4,0.2). It illustrates that the sources achieve the largest throughput under the delay-tolerant case with θ=(0,0,0), because, in this case, the UAV can fully exploit the mobility. Besides, under the fairest case, i.e., θ=(1,1,1), equal throughput performance can be achieved by each source, but the sum amount of throughput is much lower than the case of θ=(0,0,0), which means that the UAV satisfies the fairness requirement at the cost of sacrificing mobility.

In Figure 12, we illustrate the information causality for period T=60 s under θ=(1,1,1) of delay-tolerant case and λ=(1,1,1),υ=1 of delay-sensitive case. We compare the proposed design with the static UAV case where the UAV is fixed at an optimal location during the period *T*. It can be observed that the proposed design outperforms the static UAV case, as the static UAV case cannot enhance the relay’s throughput by trajectory design due to the static position, while the proposed design can enjoy the performance gains brought by UAV’s mobility. However, these two cases have some common characteristics in terms of curve tendency. Specifically, it can be observed that, at the beginning, no information transmit in the system, as the sources have to harvest energy first because they are initially assumed to be with no residual energy. When sources have enough energy to support WIT, they transmit information to the UAV in uplink until the harvested energy cannot support WIT. For the delay-tolerant case, the UAV would chose the suitable ground nodes including sources and destination to communicate as per the channel conditions, even though the factor of θ=(1,1,1) poses a fairness requirement over whole period. Different from this, the delay-sensitive case needs to severely satisfy the factor of λ=(1,1,1),υ=1 requirement in each time slot, which, as a result, causes the throughputs of S2U and U2D to be generally increasing. In addition, we can also find that the total received information from sources equals the forwarded information to destination, which is in accordance with our expectation that the equality holds in Equation (Equation 16).

In Figure 13, we plot the throughput (i.e., the information bits that the UAV/relay assists to deliver from sources to destination) for both delay-tolerant case and delay-sensitive case versus different period *T*. To illustrate our proposed design, we set the static design as a benchmark. Note that the benchmark can also be solved by our proposed Algorithm 1. Firstly, it can be observed that the delay-tolerant case outperforms the delay-sensitive case in terms of the throughput over the period *T*. This is readily comprehended as the UAV’s mobility and the scheduling are more restricted in delay-sensitive case. Next, it can be also observed that the proposed design greatly outperforms the benchmark for each case, and the case of θ=(0,0,0) is superior to the case of θ=(1,1,1), while the fairness cannot be guaranteed (e.g., see Figure 11). As a result, Figure 13 validates our proposed design as well as indicates that performance gains can be achieved by our design.

## 5. Conclusions

In this paper, we investigate UAV-enabled relaying that assists information delivery in WPCN, in which the UAV also acts as a hybrid AP to provide WPT for sources that then perform WIT to the UAV by using harvested energy. Specifically, we design the delay-tolerant case and delay-sensitive case for satisfying different communication delay demands. Our goal is to maximize the system throughput, subject to information causality and so-called neutrality constraints. By applying SCA technique, the formulated non-convex problem is effectively solved by jointly optimizing three subproblems, based on which an iterative algorithm is proposed. Finally, numerical results have shown the validation of our proposed design as well as substantial performance gains over the benchmark. 

## Figures and Tables

**Figure 1 sensors-19-02989-f001:**
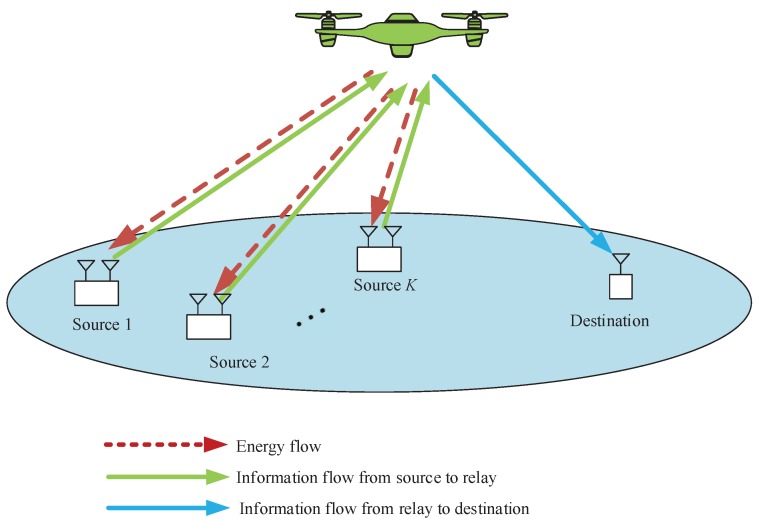
A UAV-enabled WPCN.

**Figure 2 sensors-19-02989-f002:**
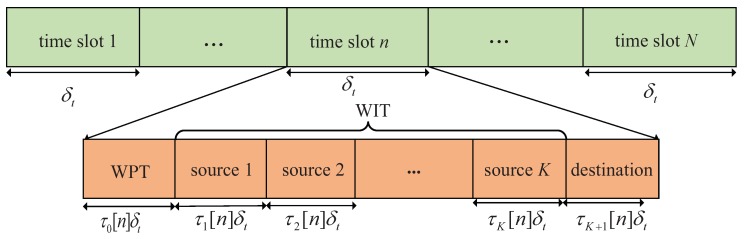
Communication protocol for the WPCN.

**Figure 3 sensors-19-02989-f003:**
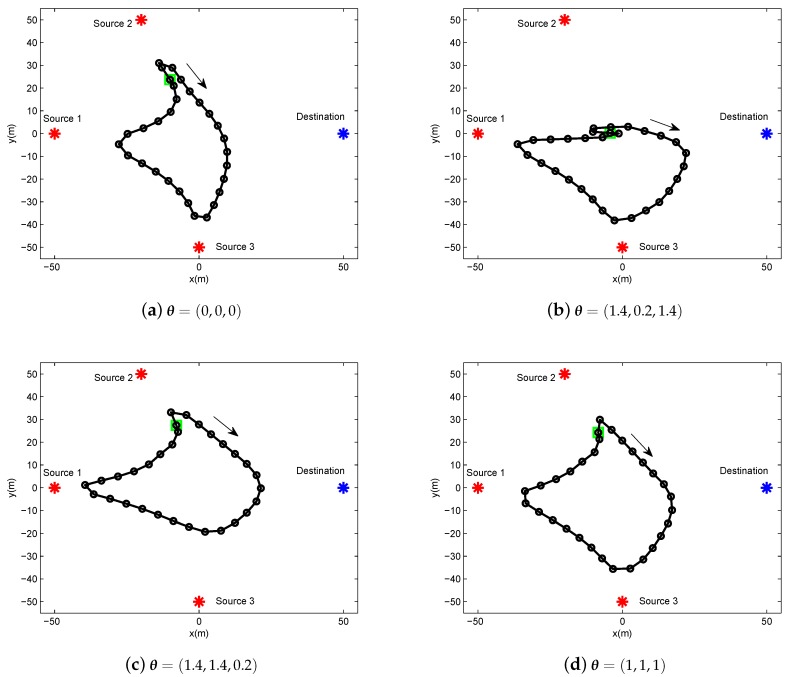
The trajectory of delay-tolerant case for period T=60 s under different factor θ.

**Figure 4 sensors-19-02989-f004:**
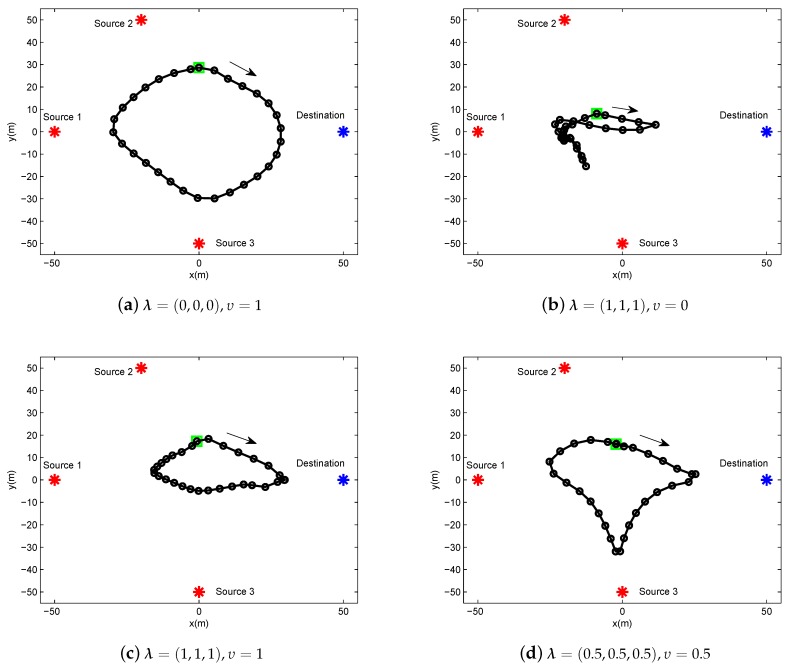
The trajectory of delay-sensitive case for period T=60 s under different factor λ,υ.

**Figure 5 sensors-19-02989-f005:**
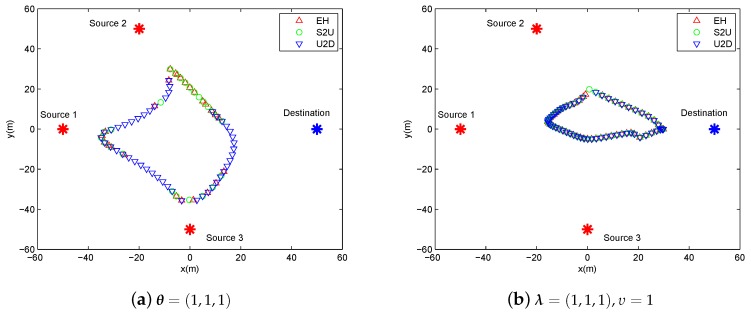
The illustration of scheduling in each slot for period T=60 s.

**Figure 6 sensors-19-02989-f006:**
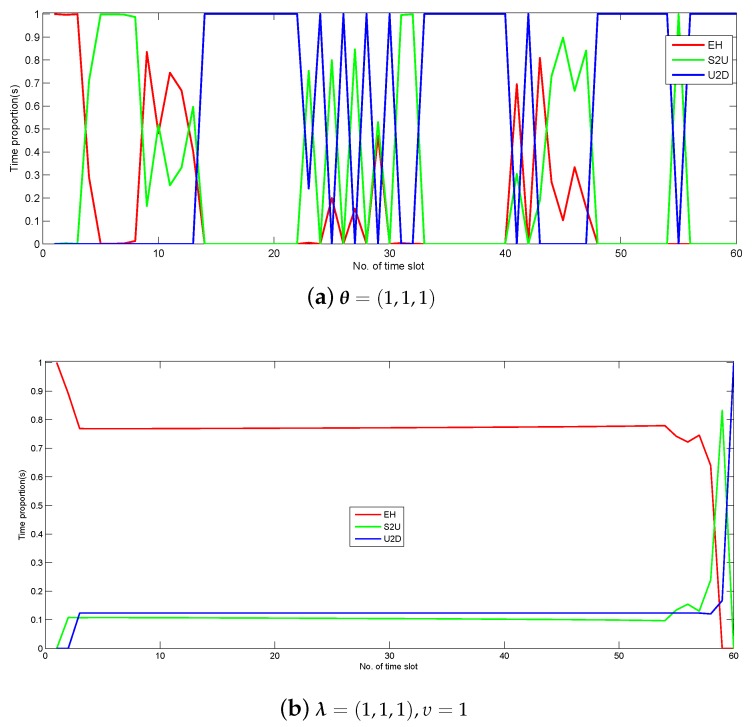
Time proportion of the scheduling in each slot under period T=60 s.

**Figure 7 sensors-19-02989-f007:**
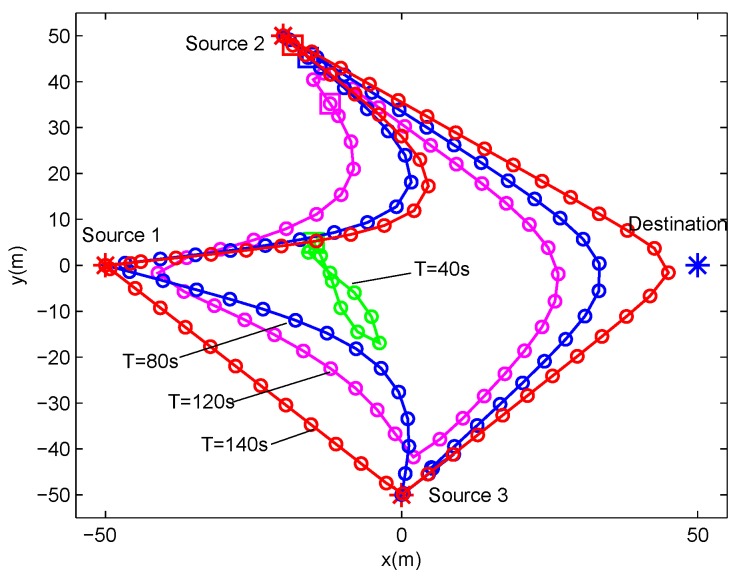
The trajectories of delay-tolerant case for different T under delay-tolerant case with θ=(0,0,0).

**Figure 8 sensors-19-02989-f008:**
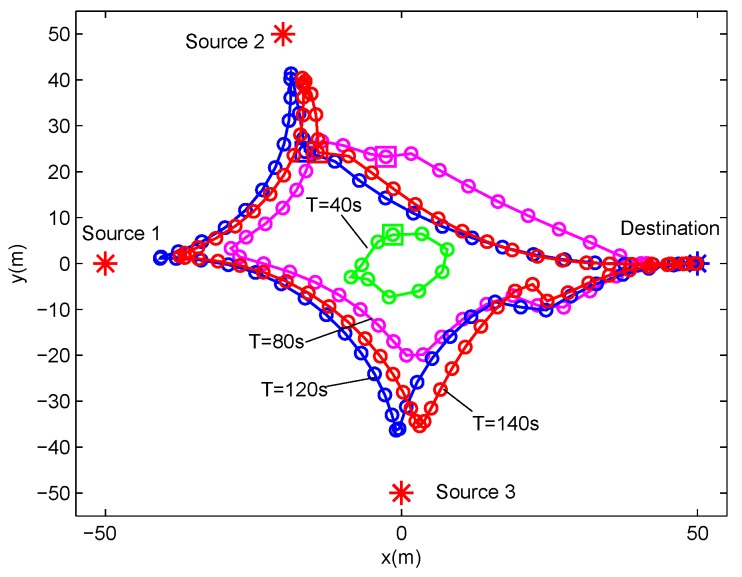
The trajectories of delay-sensitive case for different T under delay-sensitive case with λ=(1,1,1),υ=1.

**Figure 9 sensors-19-02989-f009:**
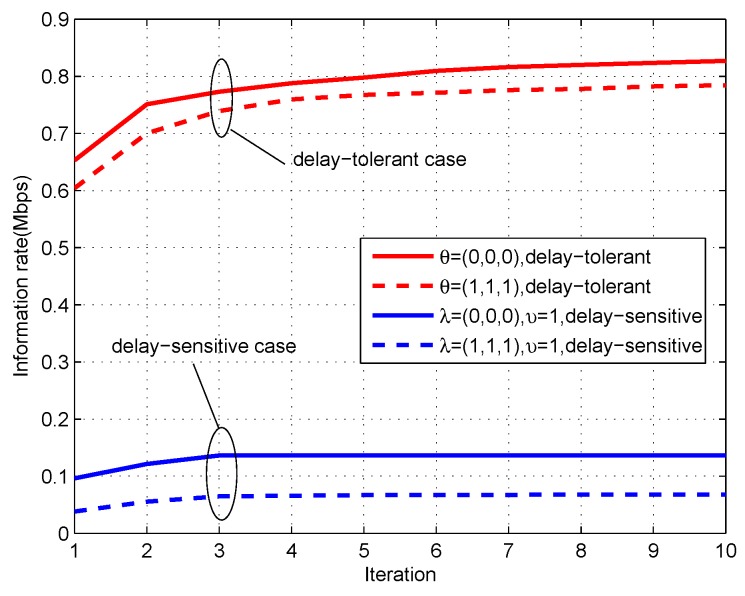
The convergence performance of the proposed algorithm for period T=60 s.

**Figure 10 sensors-19-02989-f010:**
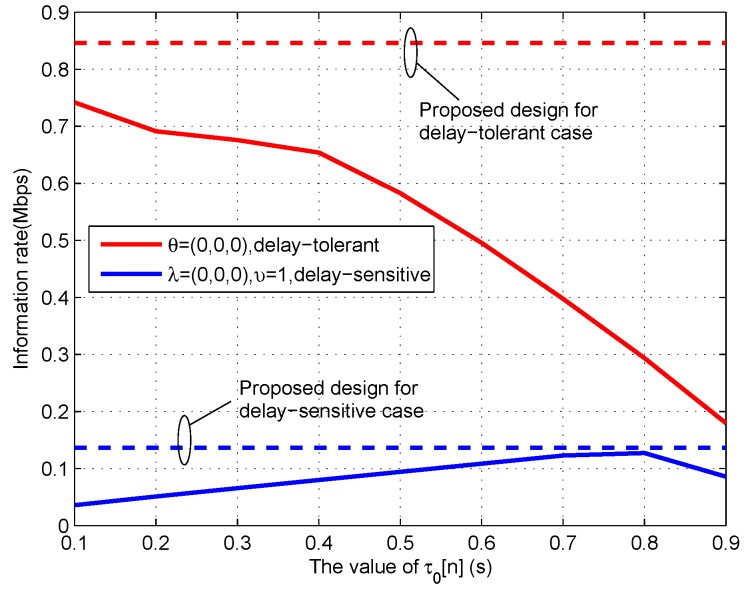
The transmit rate under fixed energy harvesting duration $\tau_{0}\left[n\right]$ for period T=60 s.

**Figure 11 sensors-19-02989-f011:**
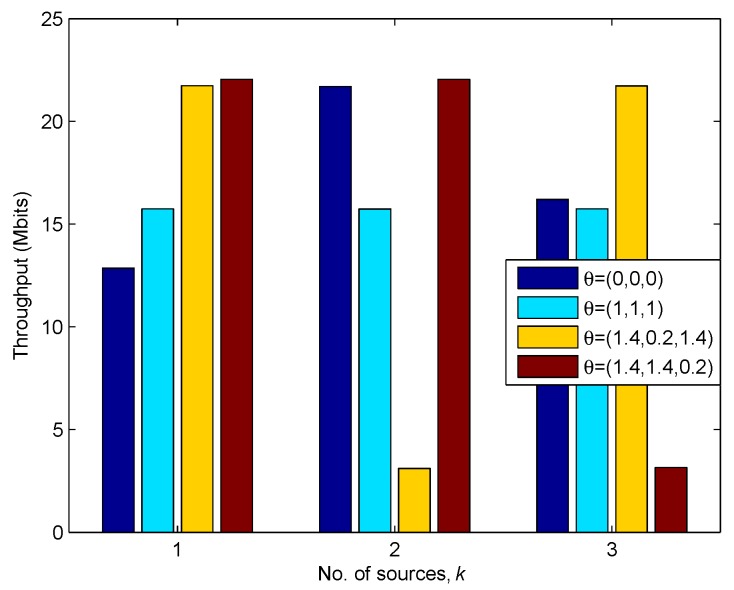
The Throughput of each source under different θ for T=60 s.

**Figure 12 sensors-19-02989-f012:**
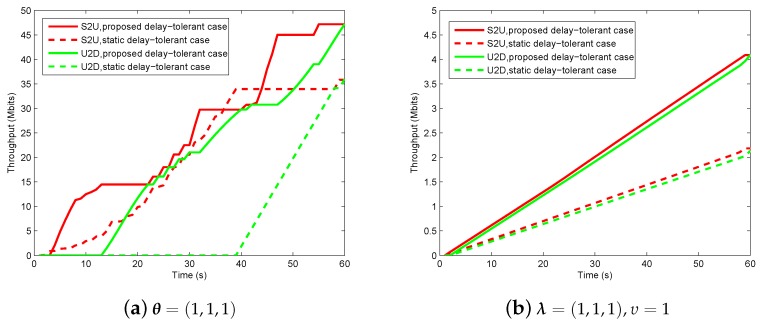
The information causality illustration for period T=60 s.

**Figure 13 sensors-19-02989-f013:**
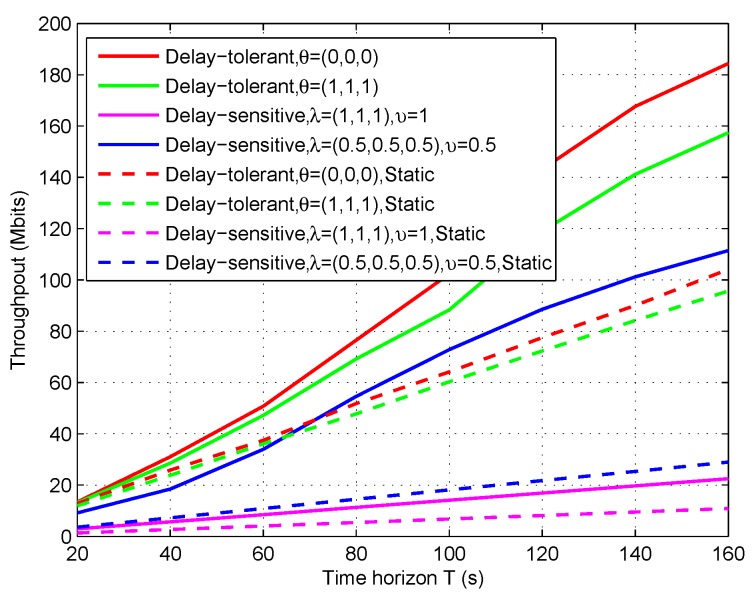
The throughput of the relaying system versus period *T*.

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
