# Peer review of "Throughput Maximization for UAV-Enabled Relaying in Wireless Powered Communication Networks"

_sensors, 2019, doi:10.3390/s19132989_

Round 1
Reviewer 1 Report
This paper studied the trajectory, power allocation and time duration optimization problem for UAV-enabled Relay in wireless power communication networks. The considered scenario is interesting and the paper is readable although there are some typo errors. The derivations seem correct although the authors adopt the existing common method. The reviewer has the following comments:
1) The authors should point out the insights from the proposed algorithm and the simulation results that should be useful for system design.
2) The analysis of the proposed algorithm should be proposed such as the convergence analysis and complexity analysis.
3) The proposed algorithm should be compared with the existing algorithms to justify the performance advantages over the existing algorithms. The exhaustive search method should also be compared.
4) There are some other closely related papers on UAV-relay:
[1] Y. Chen, W. Feng, and G. Zheng, “Optimum placement of UAV as relays,” IEEE Commun. Lett., vol. 22, no. 2, pp. 248–251, Feb 2018.
[2] R. Fan, J. Cui, S. Jin, K. Yang, and J. An, “Optimal node placement and resource allocation for UAV relaying network,” IEEE Commun. Lett.
, vol. 22, no. 4, pp. 808–811, April 2018.
[3] C. Pan, H. Ren, Y. Deng, M. Elkashlan and A. Nallanathan, "Joint Blocklength and Location Optimization for URLLC-Enabled UAV Relay Systems," IEEE Commun. Lett., vol. 23, no. 3, pp. 498-501, March 2019.
The authors should compare and emphasize the difference between your work and these papers.
5) The authors should polish this paper carefully and there are several typo errors in this paper:
(1) In (9), the comma should be period.
(2) Above (10), `Assum' should be `Assume'
(3) Above (10), delete constraints.
(4) Below (17), `solt' should be `slot'
(5) Below (P2), 'is same as' should be 'the same as'.
(6) Below (41), `can be easily verify' should be `can be easily verified that'
(7) Below (41), `can be easily verify' should be `can be easily verified that'
(8) Line 160, 'a efficient algorithm' should be 'an efficient algorithm'
Author Response
Response to Reviewer 1’s Comments on Paper “Throughput Maximization for UAV-Enabled Relaying in Wireless Powered Communication Networks” (Manuscript ID: sensors-514056) Dear reviewer, Thank you very much for your time and effort put in reviewing our manuscript. We also thank you for your valuable comments that helped improve the quality of this paper. In the following, we address your concerns in the order that they are mentioned. Comments to the Corresponding Author This paper studied the trajectory, power allocation and time duration optimization problem for UAV-enabled Relay in wireless power communication networks. The considered scenario is interesting and the paper is readable although there are some typo errors. The derivations seem correct although the authors adopt the existing common method. Response: Thank you very much for the summary and the positive comments on our work. The reviewer has the following comments: 1. The authors should point out the insights from the proposed algorithm and the simulation results that should be useful for system design. Response: Thanks for the comments. We have summarized the insights in Part 1 of the manuscript as follows: The proposed algorithm and simulation results provide helpful insights for UAV-enabled relaying design in WPCN, in which the impact of delay is considered and the UAV trajectory is optimized as well. The simulation results shed light on that in practice, the proposed algorithm contributes to achieve different delay required UAV-enabled WPCN system design. 2. The analysis of the proposed algorithm should be proposed such as the convergence analysis and complexity analysis. Response: Thanks for the comments. We have analyzed the convergence and complexity of the proposed algorithm in the revised manuscript as follows: 3.5. Convergence and Complexity Analysis of Proposed Algorithm To explain the convergence of the proposed design, we use , , and to be the objective value of problem (P1), (P1.1) and (P1.3.2), respectively, where , , . For any one iteration i >0, we can get the following expressions, (42) (43) Since is the globally optimal solution to problem (P1) by solving (P1.1) with given and . Considering problem (P1.2) always offers a lower-bounded solution to (P1), hence we can obtain (44) (45) (46) The equality (44) holds because (20) and (21) are tight bounds for (P1). Similarly, it can be easily obtained (47) (48) (49) Based on (42)-(49), we have (50) As a result, the proposed algorithm is verified to be non-decreasing in iterations. In addition, the objective value of (P1) is upper-bounded, which means that the algorithm is guaranteed to converge. In the proposed Algorithm 1, the original problem (P1) is decomposed into three subproblems that can be efficiently solved by typical method as applied in [19], [21] and [22] with low complexity. Then, these subproblems are optimized in an alternate manner. Furthermore, the optimization tool CVX is high-efficiency for solve the such convex problems [23], which makes the complexity of Algorithm 1 affordable. 3. The proposed algorithm should be compared with the existing algorithms to justify the performance advantages over the existing algorithms. The exhaustive search method should also be compared. Response: Thanks for the comments. In order to show the performance of our proposed algorithm, we followed Ref. [19] in the revised manuscript and have designed a static UAV case as the benchmark, as shown in Fig. 12. In fact, UAV trajectory design in UAV-enabled communication system is hard to tackle. In the literature, e.g. Ref. [6]-[8], [10]-[13],[19], the authors were absorbed in find a sub-optimal solution to the formulated problem, and measured the performance of the their proposed algorithms by comparing some special cases instead of other different algorithms due to the solution nature. The exhaustive search method can solve a problem with a sufficiently satisfying solution at the cost of complexity and computation time. Unfortunately, the exhaustive search method seems not to be applied to solve the formulated problem in our paper. The main reason lies on the UAV’s trajectory. Specifically, based on time discretization method, for a given period time T, the UAV has N flight status. For any time slot n∈N, assuming that the location of the UAV is q[n], the possible locations of the UAV in the next time slot n+1 is within the circle area whose center is q[n] and radius is Vmaxδt , which means that there are infinite possibilities for the next location. Furthermore, if consider the whole period time T, we have to confront infinite number of combinations of UAV location status. This is challenging even impossible for us to find the optimal solution by using exhaustive search method. 4. There are some other closely related papers on UAV-relay: [1] Y. Chen, W. Feng, and G. Zheng, “Optimum placement of UAV as relays,” IEEE Commun. Lett., vol. 22, no. 2, pp. 248–251, Feb 2018. [2] R. Fan, J. Cui, S. Jin, K. Yang, and J. An, “Optimal node placement and resource allocation for UAV relaying network,” IEEE Commun. Lett., vol. 22, no. 4, pp. 808–811, April 2018. [3] C. Pan, H. Ren, Y. Deng, M. Elkashlan and A. Nallanathan, "Joint Blocklength and Location Optimization for URLLC-Enabled UAV Relay Systems," IEEE Commun. Lett., vol. 23, no. 3, pp. 498-501, March 2019. The authors should compare and emphasize the difference between your work and these papers Response: Thanks for the comments. In the revised manuscript, the above references have been well cited in the revised manuscript. 5. The authors should polish this paper carefully and there are several typo errors in this paper: (1) In (9), the comma should be period. (2) Above (10), `Assum' should be `Assume' (3) Above (10), delete constraints. (4) Below (17), `solt' should be `slot' (5) Below (P2), 'is same as' should be 'the same as'. (6) Below (41), `can be easily verify' should be `can be easily verified that' (7) Below (41), `can be easily verify' should be `can be easily verified that' (8) Line 160, 'a efficient algorithm' should be 'an efficient algorithm' Response: Thanks for the above comments. We have corrected these errors in the revised manuscript.
Reviewer 2 Report
This paper proposes a mobile relaying in wireless communication networks using an unmanned aerial vehicle. Some important related works should be concerned, such as:
1. In Section 2, the relay transmits whole signals which the relay receives from all sources in all of previous sub time slots. Is it right? If it right, I think the relay operates on NOMA protocol not TDMA protocol.
In case of TDMA protocol, there are two hops in each sub time-slot: the first hop for information transmission between each source and the relay, and the second hop for information transmission between the relay and the destination. With other TDMA approaches, at first the relay receives signals of all sources as the authors mentions in the paper, then the relay transmits signal consecutively to the destination.
2. Please provide the proof for (16).
3. In Section 4, there are no comparison between the proposed algorithm and the existing algorithms, i.e, AF relaying.
4. Please provide the complexity of proposed algorithms.
5. To verify effectiveness of proposed algorithms, please add comparison results with the case of fixed energy harvesting duration at each time slot, tau_0 = 0.1, 0.5, 0.9,.... (s).
6. In Fig. 7, the false detection rate is presented. Please explain the detection threshold after BSM.
Author Response
Response to Reviewer 2’s Comments on Paper
“Throughput Maximization for UAV-Enabled Relaying in Wireless Powered Communication Networks”
(Manuscript ID: sensors-514056)
Dear reviewer,
Thank you very much for your time and effort put in reviewing our manuscript. We also thank you for your valuable comments that helped improve the quality of this paper. In the following, we address your concerns in the order that they are mentioned.
This reviewer has the following comments.
This paper proposes a mobile relaying in wireless communication networks using an unmanned aerial vehicle. Some important related works should be concerned, such as:
1. In Section 2, the relay transmits whole signals which the relay receives from all sources in all of previous sub time slots. Is it right? If it right, I think the relay operates on NOMA protocol not TDMA protocol.
In case of TDMA protocol, there are two hops in each sub time-slot: the first hop for information transmission between each source and the relay, and the second hop for information transmission between the relay and the destination. With other TDMA approaches, at first the relay receives signals of all sources as the authors mentions in the paper, then the relay transmits signal consecutively to the destination.
Response: Thanks for the above comments. In this paper, we design a communication protocol for the UAV relay as shown in Fig. 2 in the manuscript, and we re-plot it below.
Actually, the protocol above is designed based on TDMA. Firstly, we discretize whole period T into N time slots with equal step δt. Then, in order to collect information from sources and deliver data to destination without any interference, each time slot is divided into K+1 subslots, and the operations including WPT, information uploading for each source and information forwarding to destination occupy one dedicated subslot, respectively. Namely, in each time slot n, sources transmit its signal to the relay one by one according to the subslot partition, and relay sends its signal in the final subslot. Hence, the protocol adopted by the relay is TDMA.
In addition, what we want to explain is that in each slot, the relay may not necessarily transmit whole signals which it receives from all sources in all of previous subslots, as shown in the following figure (i.e., Fig. 12(a) of the revised manuscript).
The picture above clearly shows that within one slot n, the relay is not necessary to transmit whole signals received from all sources in all of previous subslots, because its location has an impact on the subslot scheduling. While, if we consider the whole period time T, or all time slot N, the amount of received signals equal to the amount of transmitted ones.
2. Please provide the proof for (16).
Response: Thanks for the above comments. In order to make expression (16) more easily comprehended, we have added some explanations in the revised manuscript as follows:
Here we give a brief explanation for (16). First, form above statement, we know the first and second time slot are allocated for WPT and information uploading, respectively. Namely, the information transmit for the relay starts from the third slot. In addition, due to the assumption of processing delay for the relay, resulting in that the last time slot, i.e., the N-$th$ slot, can not be used to operate WIT for sources, otherwise it would cause time resource waste. The information causality indicates that the WIT starts from the 2nd time slot, and the N-$th$ slot can only be used for information forwarding to the destination, i.e., $\tau_{K+1}[N]=1$. It can be easy to prove that, at the optimal solution, the inequality (16) would hold with equality. Otherwise, we can always increase the value of $\tau_{K+1}[j]R_{rd}[j]$ by enlarge the value of $\tau_{K+1}[j]$, which does not violate our design.
3. In Section 4, there are no comparison between the proposed algorithm and the existing algorithms, i.e, AF relaying.
Response: Thanks for the above comments. In order to show the performance of our proposed algorithm, we followed Ref. [19] in the revised manuscript and have designed a static UAV case as the benchmark, as shown in Fig. 12. In fact, UAV trajectory design in UAV-enabled communication system is hard to tackle. In the literature, e.g. Ref. [6]-[8], [10]-[13],[19], the authors were absorbed in find a sub-optimal solution to the formulated problem, and measured the performance of the their proposed algorithms by comparing some special cases instead of other different algorithms due to the solution nature.
In general, the AF relaying is two-way relay, which is significantly different from our proposed design. What’s more, adopting AF relaying may be incompatible with our algorithm design. As a result, it seems to be unsuitable to compare our DF relaying with AF relaying. In future, we are going to take UAV-enabled AF relaying as a particular problem to study.
4. Please provide the complexity of proposed algorithms.
Response: Thanks for the above comments. In the revised manuscript, we analyzed the proposed algorithm from convergence and complexity:
3.5. Convergence and Complexity Analysis of Proposed Algorithm
To explain the convergence of the proposed design, we use ,, and to be the objective value of problem (P1), (P1.1) and (P1.3.2), respectively, where , ,. For any one iteration i >0, we can get the following expressions,
(42)
(43)
Since is the globally optimal solution to problem (P1) by solving (P1.1) with given and . Considering problem (P1.2) always offers a lower-bounded solution to (P1), hence we can obtain
(44)
(45)
(46)
The equality (44) holds because (20) and (21) are tight bounds for (P1). Similarly, it can be easily obtained
(47)
(48)
(49)
Based on (42)-(49), we have
(50)
As a result, the proposed algorithm is verified to be non-decreasing in iterations. In addition, the objective value of (P1) is upper-bounded, which means that the algorithm is guaranteed to converge.
In the proposed Algorithm 1, the original problem (P1) is decomposed into three subproblems that can be efficiently solved by typical method as applied in [19], [21] and [22] with low complexity. Then, these subproblems are optimized in an alternate manner. Furthermore, the optimization tool CVX is high-efficiency for solve the such convex problems [23], which makes the complexity of Algorithm 1 affordable.
5. To verify effectiveness of proposed algorithms, please add comparison results with the case of fixed energy harvesting duration at each time slot, tau_0 = 0.1, 0.5, 0.9,.... (s).
Response: Thanks for the above comments. We have added the simulation to show the case of different fixed energy harvesting duration at each time slot in the revised manuscript. The details are also given as follows:
Fig. 10 The transmit rate under fixed energy harvesting duration $\tau_0[n]$ for period $T=60$s.
In order to show the effectiveness of the proposed algorithm 1, Fig. 10 shows the information rate received by the destination versus the fixed energy harvesting duration in each time slot, i.e., $\tau_0[n]$, for given period $T=60s$. For the convenience of comparison, we plot the achievable rate of proposed design in which the value of $\tau_0[n]$ is optimized. It can be observed from this picture that the case of fixed $\tau_0[n]$ is worse than the proposed design, and the delay-tolerant case always outperforms the delay-sensitive case. Specifically, for the delay-tolerant case of $\boldsymbol{\theta}=(0,0,0)$, the achievable rate decreases with the value of $\tau_0[n]$ increasing; while for the delay-sensitive case, the achievable rate first increases and then decreases. This phenomenon can be well comprehended by combining Fig. 6 with Fig. \10, the operation of WPT in each time slot is not the best choice for delay-tolerant case but for the delay-sensitive case. In general, with the increasing value of $\tau_0[n]$, the less time can be utilized for information transmit in each time slot, so as to result in the rate decrease for large value of $\tau_0[n]$.
6. In Fig. 7, the false detection rate is presented. Please explain the detection threshold after BSM.
Response: Thanks for the above comments. Sorry we can not understand the meaning of BSM, or maybe there is something wrong in this comment.

Round 2
Reviewer 1 Report
The authors have addressed all my previous comments. Acceptance of this paper is recommended.
Author Response
Thank you very much for the positive comments on our work.
Reviewer 2 Report
The authors answered Question 1, Question 2, Question 4 and Question 5 well. However, the reviewer has the following comment need to be addressed.
1. For Question 3, the authors need to analyze the performance of the proposed technique and static UAV in Fig 12. Please add the results of the static UAV in Fig. 12.
Author Response
Dear reviewer,
Thank you very much for your time and effort put in reviewing our manuscript. We also thank you for your valuable comments that helped improve the quality of this paper. In the following, we address your concerns in the order that they are mentioned.
The reviewer has the following comments:
The authors answered Question 1, Question 2, Question 4 and Question 5 well. However, the reviewer has the following comment need to be addressed.
1. For Question 3, the authors need to analyze the performance of the proposed technique and static UAV in Fig 12. Please add the results of the static UAV in Fig. 12.
Response: Thanks for the comments. In the revised manuscript of Round 2, we have supplemented the simulation results of static UAV case in Fig. 12, as shown below:
(a) (b)
Fig. 12 The information causality illustration for period T=60s.
What’s more, we analyze the performance of the static case and compare it with the proposed design in the revised paper in blue color. Here, we rewrite the analysis for Fig. 12 as follows:
In Fig. 12, we illustrate the information causality for period $T=60$s under $\boldsymbol{\theta}=(1,1,1)$ of delay-tolerant case and $\boldsymbol{\lambda}=(1,1,1), \upsilon=1$ of delay-sensitive case. We compare the proposed design with the static UAV case where the UAV is fixed at an optimal location during the period $T$. It can be observed that the proposed design outperforms the static UAV case, this is because that the static UAV case cannot enhance the relay's throughput by trajectory design due to the static position, while the proposed design can enjoy the performance gains brought by UAV's mobility. But these two cases have some common characteristics in terms of curve tendency. Specifically, it observes at the beginning no information transmit in the system, which means that the sources have to harvest energy at the beginning because they are initially assumed to be with no residual energy. When sources get enough energy to support WIT, they would transmit information to the UAV in uplink until the harvested energy cannot support WIT. For the delay-tolerant case, the UAV would chose the suitable ground nodes including sources and destination to communicate as per the channel conditions even though the factor of $\boldsymbol{\theta}=(1,1,1)$ poses a fairness requirement over whole period time. Different from this, the delay-sensitive case need to severely satisfy the factor of $\boldsymbol{\lambda}=(1,1,1), \upsilon=1$ requirement in each time slot, which, as a result, causes the throughputs of S2U and U2D are nearly increasing. In addition, we can also find that the total received information from sources equals the forwarded information to destination, which is in accordance with our expectation that the equality holds in (16).
